# An Individualized Low-Pneumoperitoneum-Pressure Strategy May Prevent a Reduction in Liver Perfusion during Colorectal Laparoscopic Surgery

**DOI:** 10.3390/biomedicines11030891

**Published:** 2023-03-14

**Authors:** Luis Enrique Olmedilla Arnal, Oscar Diaz Cambronero, Guido Mazzinari, José María Pérez Peña, Jaime Zorrilla Ortúzar, Marcos Rodríguez Martín, Maria Vila Montañes, Marcus J. Schultz, Lucas Rovira, Maria Pilar Argente Navarro

**Affiliations:** 1Department of Anaesthesiology, Hospital General Universitario Gregorio Marañón, 28007 Madrid, Spain; 2Perioperative Medicine Research Group, Instituto de Investigación Sanitaria La Fe, 46026 Valencia, Spain; 3Department of Anaesthesiology, Hospital Universitario y Politécnico La Fe, 46026 Valencia, Spain; 4Spanish Clinical Research Network (SCReN), SCReN-IIS La Fe, PT17/0017/0035, 46026 Valencia, Spain; 5Department of Statistics and Operations Research, University of Valencia, 46100 Valencia, Spain; 6Department of Digestive Surgery, Hospital General Universitario Gregorio Marañón, 28007 Madrid, Spain; 7Department of Intensive Care, Laboratory of Experimental Intensive Care and Anesthesiology (LEICA), Amsterdam University Medical Centers, Location ‘AMC’, 1105 AZ Amsterdam, The Netherlands; 8Mahidol Oxford Tropical Medicine Research Unit (MORU), Mahidol University, Bangkok 10400, Thailand; 9Nuffield Department of Medicine, University of Oxford, Oxford OX3 7BN, UK; 10Department of Anaesthesiology, Consorcio Hospital General Universitario de Valencia, 46014 Valencia, Spain

**Keywords:** laparoscopy, colorectal surgery, laparoscopic colorectal surgery, pneumoperitoneum, perioperative medicine, low-impact laparoscopy

## Abstract

High intra-abdominal pressure (IAP) during laparoscopic surgery is associated with reduced splanchnic blood flow. It is uncertain whether a low IAP prevents this reduction. We assessed the effect of an individualized low-pneumoperitoneum-pressure strategy on liver perfusion. This was a single-center substudy of the multicenter ‘Individualized Pneumoperitoneum Pressure in Colorectal Laparoscopic Surgery versus Standard Therapy II study’ (IPPCollapse–II), a randomized clinical trial in which patients received an individualized low-pneumoperitoneum strategy (IPP) or a standard pneumoperitoneum strategy (SPP). Liver perfusion was indirectly assessed by the indocyanine green plasma disappearance rate (ICG–PDR) and the secondary endpoint was ICG retention rate after 15 min (R_15_) using pulse spectrophotometry. Multivariable beta regression was used to assess the association between group assignment and ICG–PDR and ICG–R_15_. All 29 patients from the participating center were included. Median IAP was 8 (25th–75th percentile: 8–10) versus 12 (12,12) mmHg, in IPP and SPP patients, respectively (*p* < 0.001). ICG–PDR was higher (OR 1.42, 95%-CI 1.10–1.82; *p* = 0.006) and PDR–R_15_ was lower in IPP patients compared with SPP patients (OR 0.46, 95%-CI 0.29–0.73; *p* = 0.001). During laparoscopic colorectal surgery, an individualized low pneumoperitoneum may prevent a reduction in liver perfusion.

## 1. Introduction

Intra-abdominal (IAP) levels higher than 15 mmHg have been shown to reduce splanchnic blood flow in animal models [1] and in humans undergoing laparoscopic cholecystectomies [2]. IAP is usually set at fixed high levels during laparoscopic surgery, and it is uncertain if an individualized low-pneumoperitoneum-pressure strategy can prevent a reduction in abdominal blood perfusion [3]. Indocyanine green (ICG) is a dye almost exclusively extracted by the liver at a very high rate upon injection, which is eliminated in an unmetabolized state through unconjugated biliary excretion without enterohepatic recirculation. Its spectrophotometric determination is independent of oxygen saturation and bilirubin concentration and is regarded as a proxy for both hepatic blood flow and function [4]. As part of a randomized trial comparing an individualized low-pneumoperitoneum-pressure strategy with a fixed high-pneumoperitoneum-pressure one [5], we compared measures of ICG clearance. It was hypothesized that an individualized low pneumoperitoneum pressure would prevent a reduction in liver blood flow.

## 2. Materials and Methods

This was a single-center pre-planned substudy of the ‘Individualized Pneumoperitoneum Pressure in Colorectal Laparoscopic Surgery versus Standard Therapy II study’ (IPPCollapse–II), a randomized clinical trial in patients undergoing laparoscopic colorectal surgery. Study details [6] and primary results have been reported elsewhere [5]. The Institutional Review Board of the Hospital Universitario y Politécnico la Fe in Valencia, Spain, approved the study protocol as well as a subsequent modification of the protocol concerning extension of recruitment. The study protocol and study conduct were in compliance with the Helsinki Declaration and Spanish legislation for biomedical research. Written informed consent was obtained from all subjects before entering the trial. The study was registered before patient enrolment at EudraCT (study identifier: 2016–001693–15) and clinicaltrials.gov (study identifier NCT03435913), and the study protocol was prepublished and updated. Patients included either received an individualized low pneumoperitoneum strategy (IPP) in which IAP was titrated down to a level at which the surgeon had a sufficient surgical working space with a minimum of 8 mmHg, or a standard pneumoperitoneum strategy (SPP) consisting of a fixed IAP at 12 mmHg. With both strategies, the surgeon could request a higher IAP up to 15 mmHg if needed.

We used the indocyanine green (ICG) blood clearance as a proxy for liver perfusion [7]. A sterile water dilution of ICG monosodium salt with a concentration of 5 mg∙mL^−1^ was prepared. The injection was carried out 15 min after pneumoperitoneum establishment according to group randomization if hemodynamic status was deemed stable. After intravenous injection of 0.5 mg∙kg^−1^ of the dye solution, the plasma disappearance rate (ICG–PDR) and the retention rate after 15 min (R_15_) were determined using a LIMON Monitor (Pulsion Medical Systems AG, Munich, Germany) fifteen minutes after establishing pneumoperitoneum. The LIMON monitor measures ICG plasma concentrations noninvasively by pulse spectrophotometry with a finger clip sensor that detects four near-infrared wavelengths [8]. PDR values are expressed as the percentage of ICG of the initial value per minute (%/min) and R_15_ values as the percentage of ICG of the initial value remaining 15 min after the injection. The primary endpoint was ICG–PDR expressed as a percentage change over time with the initial value set as 100%. The secondary endpoint was ICG–R_15_. Values of ICG−PDR of >18% and of 6–12% ICG–R_15_ are considered to be normal [4].

We had no a priori sample size calculation but used data from all patients included in the Hospital Gregorio Marañon, Madrid, Spain. Continuous variables are reported as median [25th–75th percentile]. Normality was checked by quantile–quantile plot examination. Categorical variables are reported as percentages and proportions. To assess the association of the two pneumoperitoneum strategies with ICG–PDR and ICG–R_15_, we estimated a multivariable beta regression introducing age, body mass index (BMI), ASA risk score, gender, and rectal versus colon surgery as covariables for each main dependent variable. In one posthoc analysis, we assessed the association between IAP and ICG–PDR and ICG–R_15_ fitting a generalized additive model (GAM) with a beta distribution for exploratory purposes. All analyses were carried out with R software version 4.0.2. Two-tailed *p* < 0.050 was considered statistically significant and no correction for multiple comparisons was performed.

## 3. Results

The substudy included 29 patients. Baseline characteristics are shown in Table 1. Median surgery time was 135 min, rectal surgery was performed more frequently in IPP patients, and patients in the SPP group were more often female. The ICG–PDR was higher in IPP patients compared with SPP patients (OR 1.42, 95%-CI 1.10–1.82; *p* = 0.006, Figure 1 and Appendix A). The PDR–R_15_ was lower in IPP patients compared with SPP patients (OR 0.46, 95%-CI 0.29–0.73; *p* = 0.006, Figure 1 and Appendix A). A higher ASA score was associated with lower ICG–PDR values and higher PDR–R_15_ values. 

## 4. Discussion

The findings of this study can be summarized as follows: (1) an IPP strategy prevented a reduced ICG–PDR; and (2) patients receiving an IPP strategy had lower PDR–R_15_; (3) of all patient characteristics, only the ASA score was associated with ICG–PDR and PDR–R_15_. Together, these findings suggest that an IPP strategy may prevent the reduction in liver perfusion during colorectal laparoscopic surgery.

The strengths of this study are its single-center design, thereby minimizing differences in care and expertise, strict inclusion and exclusion criteria leading to homogenous study groups, the prespecified and strictly followed analysis plan, and the multivariable analysis to account for potential confounding factors.

Despite the current recommendations [9], IAP is usually fixed and high during laparoscopic abdominal surgery, with little attention paid to the marginal gains in the surgical working space with IAP increments. However, a renewed interest in the IAP levels used during laparoscopic surgery has recently surged, and its potential injurious effect is increasingly recognized. The findings of the current study add to this and are in line with the findings of a previous study that showed IAP to decrease peritoneal perfusion [3]. Indeed, the effect of increased IAP is well documented in studies carried out in the Intensive Care Unit in patients with intraabdominal hypertension. Levels of IAP and ICG–PDR correlated in this population and levels of IAP of >15 mmHg were more associated with impaired ICG clearance values [10,11].

Since the population studied herein did not have hepatic dysfunction or biliary obstruction, we consider the reduction in ICG–PDR to reflect a reduction in liver perfusion.

Performing laparoscopic surgery at lower IAP levels is feasible [12] and associated with less postoperative pain [13]. Fixed and semi-fixed boundaries delimit the abdomen; therefore, a pressure increment expands its volume up to a point according to its biomechanics. This is associated with numerous effects both in the abdomen microenvironment and with increased ocular and intrathoracic pressure and decreased cardiac output [13,14,15].

We acknowledge some limitations of this study. First, its single-center design and the strict inclusion criteria reduce the generalizability of the findings. The open-label design and small sample size reduce external validity. We did not assess hepatic blood flow directly and did not assess the effects on microcirculation; we also did not collect liver function blood tests, although underlying hepatocellular dysfunction is unlikely since patients with a known hepatic disease were excluded as per the trial protocol. Data on previous chemotherapy were not available thus a modifying effect could not be tested. Moreover, we did not collect central venous pressure nor preload data, although the fluids were administered in the same fashion in both groups and the measurements were carried out at the beginning stages of the surgery; thus, a modifying effect from different preload conditions is unlikely. Since we measured the ICG–PDR at only one time point, we remain uncertain about dynamic changes associated with an IPP strategy. Nevertheless, as we measured the ICG clearance when the IAP was stabilized at its minimum level, a change is unlikely.

In conclusion, during laparoscopic colorectal surgery, an individualized low pneumoperitoneum strategy, compared with a standard pneumoperitoneum strategy, may prevent a reduction in liver perfusion. Future studies should use other markers of liver perfusion and are needed to understand the effect of a lower IAP strategy on patient-centerd outcomes.

## Figures and Tables

**Figure 1 biomedicines-11-00891-f001:**
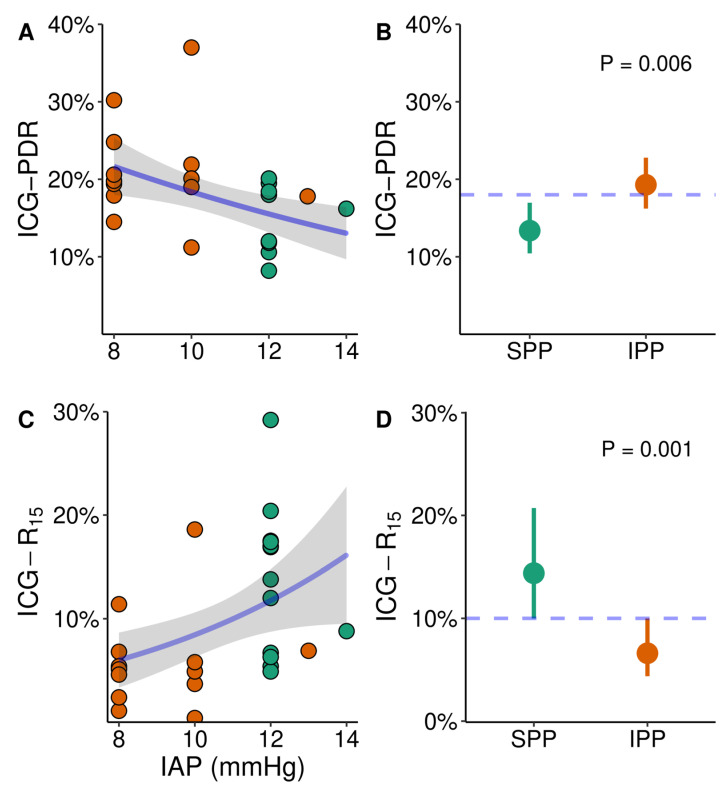
(**A**) Generalized additive beta regression model estimation of IAP’s effect on ICG–PDR. (**B**) Marginal effect of group assignment, i.e., IPP versus SPP, on ICG–PDR as estimated by the multivariable beta regression model. Values above the dashed blue line are considered normal. (**C**) Generalized additive beta regression model estimation of IAP’s effect on ICG–R_15_. Continuous blue lines are the effect of IAP as estimated by each model, and the gray shadows are the 95% CI bandwidth. (**D**) Marginal effect of group assignment, i.e., IPP versus SPP, on ICG–R_15_ as estimated by the multivariable beta regression model. Values below the dashed blue line line are considered normal. In all panels, orange represents the IPP group and green the SPP group. Abbreviations: ICG, Indocyanine green. PDR, plasma disappearance rate. R_15_, retention rate after fifteen minutes. IPP, individualized pneumoperitoneum pressure. SPP, standardized pneumoperitoneum pressure.

**Table 1 biomedicines-11-00891-t001:** Patients’ characteristics.

	All (*n* = 29)	SPP Group (*n* = 14)	IPP Group (*n* = 15)	SMD
**Age** (years)	66 (60–74)	71 (63–77)	64 (58–69)	0.61
**Sex** (female)	5 (17.2%)	5 (35.7%)	0 (0.0%)	1.05
**ASA risk score**				0.26
I	3 (10.3%)	1 (7.1%)	2 (13.3%)	
II	19 (65.5%)	10 (71.4%)	9 (60.0%)	
III	7 (24.1%)	3 (21.4%)	4 (26.7%)	
**Intraoperative IAP** (mmHg)				5.32
8	8 (27.6%)	0 (0.0%)	8 (53.3%)	
10	5 (17.2%)	0 (0.0%)	5 (33.3%)	
12	13 (44.8%)	12 (85.7%)	1 (6.7%)	
13	1 (3.4%%)	0 (0.0%)	1 (6.7%)	
14	2 (6.9%)	2 (14.3%)	0 (0.0%)	
**BMI** (Kg · m^−2^)	26.8 (3.6)	26.5 (3.6)	27.1 (3.7)	0.16
**Type of surgery** (indication)				0.80
Subtotal colectomy	1 (3.4%)	1 (7.1%)	0 (0.0%)	
Total colectomy	2 (6.9%)	1 (7.1%)	1 (6.7%)	
Right hemicolectomy	9 (31.0%)	6 (42.9%)	3 (20.0%)	
Left hemicolectomy	1 (3.4%)	0 (0.0%)	1 (6.7%)	
Sigmoidectomy	5 (17.2%)	2 (14.3%)	3 (20.0%)	
Rectal anterior resection	11 (37.9%)	4 (28.6%)	7 (46.7%)	
**Type of surgery** (Rectal)	11 (37.9%)	4 (28.6%)	7 (46.7%)	0.38
**Surgery duration** (minutes)	135 (105–210)	135 (104–201)	135 (110–212)	0.07
**ICG–PDR** (%)	18% (6%)	14% (4%)	20% (7%)	1.14
**ICG–R_15_** (%)	10% (7%)	13% (7%)	7% (5%)	1.04

Data are reported as mean (SD) or median [25th to 75th percentile] or *n* (%). Abbreviations: SMD, standardized mean difference; ASA, American Society of Anesthesiology, BMI, Body Mass Index; SPP, standard pneumoperitoneum pressure; IPP, individualized pneumoperitoneum pressure; IAP, intra-abdominal pressure; ICG, indocyanine green; PDR, plasma disappearance rate; R_15_, retention rate after 15 min.

## Data Availability

The principal investigator, Oscar Diaz Cambronero, had full access to all the data in the study and took responsibility for the data’s integrity and the data analysis’s accuracy. The full protocol and the results for the main study have been published previously (Trials 2019; 20:190 & 2020; 21:70, British Journal of Surgery; 7 June 2020. https://doi.org/10.1002/bjs.11736).

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
