# Peer review of "An Individualized Low-Pneumoperitoneum-Pressure Strategy May Prevent a Reduction in Liver Perfusion during Colorectal Laparoscopic Surgery"

_biomedicines, 2023, doi:10.3390/biomedicines11030891_

Round 1

Reviewer 1 Report

This is a valuable paper suggesting the effect of low intraabdominal pressure during laparoscopic surgery to prevent the risk of reduction in liver perfusion. The major concern is, as the authors described, whether ICG-PDR accurately reflected live perfusion. I have a few comments for the revision of this paper as the following.

1. In the introduction, the authors are expected to present how the ICG can appropriately reprersent liver perfusion.

2. In the method, please mention when the ICG was injected.

3. Although the authors described that they did not collect data on liver function blood test, I think data such as the variables defining Child-Pugh score, total cholesterol, or cholinesterase can be easily collected. In addition, the history of chemotherapy should be presented because extensive chemotherapy often cause hepatic injury. Especially oxaliplatin induced sinusoidal injury would affect hepatic perfusion.

4. Intraoperative amount of fluid infusion or central venous pressure would also affect hepatic perfusion. Can the authors present these data?

Author Response

  1. In the introduction, the authors are expected to present how the ICG can appropriately represent liver perfusion.

We agree. Thank you for pointing this out. We added an explanation of indocyanine green and its measurement for hepatic function and blood flow in the introduction of the updated version of the manuscript which reads as follows:

“Indocyanine green (ICG) is a dye almost exclusively extracted by the liver at a very high rate upon injection. It is eliminated unmetabolized through unconjugated biliary excretion without enterohepatic recirculation. Its spectrophotometric determination is independent of oxygen saturation and bilirubin concentration and is regarded as a proxy for both hepatic blood flow and function [4].” (Page 1, Line 46)

  1. In the method, please mention when the ICG was injected.

Thank you for pointing this out. We clearly report when ICG was injected in the updated version of the manuscript as follows:

“The injection was carried out 15 minutes after pneumoperitoneum establishment according to group randomization if hemodynamic status was deemed stable.” (Page 2, Line 74)

  1. Although the authors described that they did not collect data on liver function blood test, I think data such as the variables defining Child-Pugh score, total cholesterol, or cholinesterase can be easily collected. In addition, the history of chemotherapy should be presented because extensive chemotherapy often causes hepatic injury. Especially oxaliplatin-induced sinusoidal injury would affect hepatic perfusion.

Thank you for the interesting remark. As we reported in the published protocol (Trials (2019) 20:190. https://doi.org/10.1186/s13063-019-3255-1), a known hepatic disease was an exclusion criterion, therefore we can reasonably exclude that a potential hepatocellular dysfunction or biliary obstruction, whether related to chemotherapy or not, could act as an unaccounted confounder on our results. Nevertheless, we introduced the ASA risk score in our multivariable model precisely to control for the physical status of patients which can be related to potential underlying physiologic impairment. We acknowledge that we could not report quantitative test results such as cholinesterase or type of preoperative chemotherapy. We further report on this limitation and our control for confounding strategy in the discussion section of the updated manuscript as follows:

“We did not assess hepatic blood flow directly and did not assess the effects on micro-circulation; we also did not collect liver function blood tests, although an underlying hepatocellular dysfunction is unlikely since patients with a known hepatic disease were excluded as per the trial protocol. Data on previous chemotherapy was not available thus a modifying effect cannot be tested.” (Page 5, Line 149)

  1. Intraoperative amount of fluid infusion or central venous pressure would also affect hepatic perfusion. Can the authors present these data?

Please see also our reply to comment 2 from this reviewer. Fluid infusion was administered according to the Spanish ERAS guidelines in both groups as reported in the published protocol and the paper describing the main study and the dye injection was carried out in stable hemodynamic conditions. Central venous catheter placement was not required by our study protocol, thus unfortunately we cannot report data on central venous pressure. We clearly state this limitation in the updated version of the discussion as follows:

“Moreover, we did not collect central venous pressure nor preload data although the fluids were administered in the same fashion in both groups and the measurement was carried out at the beginning stages of the surgery thus a modifying effect from different preload conditions is unlikely.” (Page 5, Line 153)

Reviewer 2 Report

The introduction needs substantial improvement. 

Despite being an old concern in advanced laparoscopic surgery, mesenteric perfusion can now be assessed using ICG. My only concern is if the results of the present paper are included in the master study. 

Author Response

  1. The introduction needs substantial improvement. 

Please see also our reply to comment 1 from reviewer 1. We reworked the introduction of the paper, and the updated version now reads as follows:

“Intra–abdominal (IAP) levels higher than 15 mmHg have been shown to reduce splanchnic blood flow in animal models [1], and in humans undergoing laparoscopic cholecystectomies [2]. IAP is usually set at fixed high levels during laparoscopic surgery, and it is uncertain if an individualized low pneumoperitoneum pressure strategy can prevent a reduction in abdominal blood perfusion [3]. Indocyanine green (ICG) is a dye almost exclusively extracted by the liver at a very high rate upon injection which is eliminated unmetabolized through unconjugated biliary excretion without enterohepatic recirculation. Its spectrophotometric determination is independent of oxygen saturation and bilirubin concentration and is regarded as a proxy for both hepatic blood flow and function [4]. As part of a randomized trial comparing an individualized low pneumoperitoneum pressure strategy with a fixed high pneumoperitoneum pressure one [5], we compared measures of ICG clearance. It was hypothesized that an individualized low pneumoperitoneum pressure would prevent a reduction in liver blood flow.” (Page 1, Line 42)

  1. Despite being an old concern in advanced laparoscopic surgery, mesenteric perfusion can now be assessed using ICG. My only concern is if the results of the present paper are included in the master study. 

We did not report data on this substudy on the published paper reporting the results of the main study as can be checked at open access British Journal of Surgery 2020 Nov;107(12):1605-1614. doi: https://doi.org/10.1002/bjs.11736.

Reviewer 3 Report

The idea of the reviewed communication is interesting and could improve results in clinical practice. However, some elements could be improved before publication:

1. The Results: The table with general data should precede the graphics. Some more explanations could be interesting for the readers regarding the perfusion measurements.

2. The Discussions: please add a paragraph with comparative results of other similar studies

Author Response

  1. The Results: The table with general data should precede the graphics. Some more explanations could be interesting for the readers regarding the perfusion measurements.

Thank you. We fixed the order of the table and graph. Moreover, we added an explanation of how to interpret such perfusion measurements in the methods section as follows:

“PDR values are expressed as the percentage of ICG of the initial value per minute (%/min), and R15 values as the percentage of ICG of the initial value remaining 15 minutes after the injection. The primary endpoint was ICG–PDR expressed as a percentage change over time with the initial value set as 100%. The secondary endpoint was ICG–R15. Values of ICG–PDR of > 18% and of 6–12 ICG–R15 are considered to be normal [4].” (Page 2, Line  81)

  1. The Discussions: please add a paragraph with comparative results of other similar studies

We added a paragraph to discuss further the implication of intraabdominal hypertension and its correlation with ICG clearance parameters in the discussion section. The paragraph now reads as follows:

“The findings of the current study add to this and are in line with a previous study that showed IAP to decrease peritoneal perfusion [3]. Indeed, the effect of increased IAP is well documented in studies carried out in the Intensive Care Unit in patients with intraabdominal hypertension. Levels of IAP and ICG-PDR correlated in this population and levels of IAP of > 15 mmHg were more associated with impaired ICG clearance values [10, 11].” (Page 5, Line 134)

Round 2

Reviewer 1 Report

The authors adequately responded to reviewers' comments.  I have no additional comments for further revision.

Reviewer 3 Report

The authors have made the required changes.